# Lactic Acid Bacteria (LAB) and Neuroprotection, What Is New? An Up-To-Date Systematic Review

**DOI:** 10.3390/ph16050712

**Published:** 2023-05-07

**Authors:** Nurliana Abd Mutalib, Sharifah Aminah Syed Mohamad, Nor Atiqah Jusril, Nur Intan Hasbullah, Mohd Cairul Iqbal Mohd Amin, Nor Hadiani Ismail

**Affiliations:** 1Atta-ur-Rahman Institute for Natural Product Discovery, Universiti Teknologi MARA Cawangan Selangor, Puncak Alam 42300, Selangor, Malaysia; 2Faculty of Applied Sciences, Universiti Teknologi MARA, Shah Alam 40450, Selangor, Malaysia; 3Faculty Bioresources and Food Industry, Universiti Sultan Zainal Abidin, Besut Campus, Besut 22200, Terengganu, Malaysia; 4Faculty of Applied Sciences, Universiti Teknologi MARA, Cawangan Negeri Sembilan, Kampus Kuala Pilah, Kuala Pilah 72000, Negeri Sembilan, Malaysia; 5Centre for Drug Delivery Technology, Faculty of Pharmacy, Universiti Kebangsaan Malaysia, Jalan Raja Muda Abdul Aziz, Kuala Lumpur 50300, Selangor, Malaysia

**Keywords:** lactic acid bacteria, LAB, probiotics, neuroprotection, neurodegenerative diseases, Alzheimer’s, Parkinson’s

## Abstract

Background: In recent years, the potential role of probiotics has become prominent in the discoveries of neurotherapy against neurodegenerative diseases, such as Alzheimer’s and Parkinson’s diseases. Lactic acid bacteria (LAB) exhibit neuroprotective properties and exert their effects via various mechanisms of actions. This review aimed to evaluate the effects of LAB on neuroprotection reported in the literature. Methods: A database search on Google Scholar, PubMed, and Science Direct revealed a total of 467 references, of which 25 were included in this review based on inclusion criteria which comprises 7 in vitro, 16 in vivo, and 2 clinical studies. Results: From the studies, LAB treatment alone or in probiotics formulations demonstrated significant neuroprotective activities. In animals and humans, LAB probiotics supplementation has improved memory and cognitive performance mainly via antioxidant and anti-inflammatory pathways. Conclusions: Despite promising findings, due to limited studies available in the literature, further studies still need to be explored regarding synergistic effects, efficacy, and optimum dosage of LAB oral bacteriotherapy as treatment or prevention against neurodegenerative diseases.

## 1. Introduction

With the rising life expectancy in most nations, neurological disorders are predicted to become more prominent. Dementia presently affects about 50 million individuals and by 2050, that number is expected to rise to 130 million [1]. The term “neurodegenerations” refers to a broad category of diseases with a common clinical presentation. These diseases often combine cognitive decline with movement problems, particularly parkinsonism or motor neuron diseases [2]. Neurodegenerative diseases (ND) are a pathologically, clinically, and genetically diverse group of disorders characterized by progressive deterioration of glia or neurons, their network, and synapses connection [3]. The most common risk factor for ND is aging. Still, there is a significant difference between the cognitive, structural, and neurometabolic changes caused by healthy aging and those that emerge from ND [4]. The neurodegeneration process involves molecular and cellular mechanisms, such as neuroinflammation, abnormal protein aggregation and misfolding, mitochondrial dysfunction, oxidative stress, reduction in neurotransmitter levels, and metal dyshomeostasis [5].

Although there was a lot of research that emphasized the pathogenesis of ND and potential treatments, the currently available therapies are limited and primarily targeting symptoms rather than the mechanisms underlying the pathology. Cholinesterase inhibitors and the N-methyl-D-aspartate receptor antagonist memantine are two approved medications that represent the standard of care for many Alzheimer’s disease (AD) patients. However, since the Seminar in 2016, no additional symptomatic cognitive enhancement drug has received global approval [6]. Apart from drugs, natural products are recognized as promising alternatives for treating neurodegeneration because of their wide range of pharmacological and biological actions and their potential to aid in pharmaceutical development and discovery. The utilization of natural products as prospective treatments for neurodegeneration was shown in several trials to have health-promoting benefits [7]. Some examples of medicinal plants and natural products that exert neuroprotective activities include *Centella asiatica* [8], *Astragalus membranaceus* [9], *Panax ginseng* [10], quercetin [11], and gallic acid [12,13].

Different interventions have been devoted to improve biological outcomes in neurological conditions, which include the use of probiotics [14]. Probiotics are defined by the World Health Organization (WHO) as “live microorganisms which when administered in adequate amounts confer a health benefit on the host” [15]. This definition implies that the live microorganisms must demonstrate beneficial health effects in human hosts to be considered as probiotics; thus, some lactic acid bacteria (LAB) are probiotics but not all [16]. LAB are Gram-positive bacteria that utilize carbohydrates as the sole source of carbon [17]. These nonsporulating, anerobic or facultative aerobic, with cocci- or rod-shaped bacteria, produce lactic acid as the major fermentation product from the metabolism of carbohydrate [18].

Historically, the health benefits of LAB include enhancing the digestion of nutrients, reducing the symptoms of lactose intolerance, and generating functional substances in the gastrointestinal tract [19]. It was previously found that a diet consisting of weekly consumption of tofu and fermented soybean product, tempeh, was positively associated with immediate memory recall, which was significantly demonstrated in elderlies with an average age of 67 years in Central Java, Indonesia [20]. A recent study in the Japanese population has revealed that the intake of natto, a fermented soybean product may contribute to reducing the risk of disabling dementia in women, especially in those aged under 60 years [21]. Presently, more and more studies have been focusing on the impact of gut microbiota towards neurological health [22]. Neuroprotection refers to the strategies and mechanisms used to guard the central nervous system (CNS) against injury due to both chronic ND, such as dementia, Parkinson’s, Alzheimer’s, and epilepsy, and acute ND, such as trauma or stroke [23]. Considerable interest has been devoted in recent years to unravel the neuroprotective role of gut microbiota because plenty of studies have described bidirectional communication between the gut and the CNS [24]. It was reported recently that gut dysbiosis is an established key player in the pathogenesis of age-related neurodegenerations [25]. Apparently, the potential role of oral bacteriotherapy is emerging in the treatment of CNS-related pathologies [26]. Therefore, this systematic review aimed to evaluate and critically discuss the potential role of LAB as preventive and therapeutic approaches in neuroprotection based on recent evidence from in vitro, in vivo, and clinical studies.

## 2. Results and Discussion

### 2.1. Database Search

From the initial primary database search, a total of 468 records were identified, of which 465 entries remained after the removal of duplicates (Figure 1). After filtering the 465 titles, 436 were excluded and 29 were identified as relevant for further review and the full text was retrieved. Following a full-text review, 26 relevant entries were selected, and three were excluded due to reasons outlined in the selection criteria. Out of 26 selected studies, 7 were in vitro studies (Table 1), 17 were in vivo or animal studies (Table 2), and 2 were clinical studies (Table 3).

### 2.2. Lactic Acid Bacteria (LAB)

LAB are classified according to their cellular morphology, growth temperature range, glucose fermentation mode, and patterns of sugar utilization [27]. In the phylum *Firmicutes*, class *Bacilli*, and order *Latobacillales*, LAB are further classified into numerous genera [28]. The genera of LAB include cocci *Aerococcus*, *Alloiococcus*, *Enterococcus*, *Lactococcus*, *Leuconostoc*, *Oenococcus*, *Pediococcus*, *Streptococcus*, *Tetragenococcus*, and *Vagococcus*; rods *Carnobacterium*, *Dolosigranulum*, and *Lactobacillus*; and the rod or coccoid-shaped genus *Weissella* [29]. The largest genus among LAB is genus *Lactobacillus*, which consists of over 237 species, with the continuous discovery of new species, such as *Lactobacillus timonensis* and *Lactobacillus metriopterae* [30]. This group of bacteria is widely spread in nature and commonly isolated from fermented dairy, meat and vegetable, the gastrointestinal and urogenital tracts of humans and animals, as well as soil and water sources [31]. Without the ability to synthesize cytochrome or other enzymes with porphyrin ring (heme), LAB cannot generate energy via proton gradient; instead, LAB utilize fermentative metabolism to yield energy [32]. Therefore, carbohydrate-rich environments are ideal for LAB growth or proliferation [33]. LAB are microorganisms that are Generally Recognized as Safe (GRAS) by the United States Food and Drug Administration (FDA), as well as Qualified Presumption of Safety (QPS) by the European Food Safety Authority (EFSA) [34]. LAB strains that have the ability to survive the acidic environment of the stomach as well as the exposure to bile salts, adhere to the intestinal mucosa, and exhibit inhibitory effect towards pathogenic bacteria, may have the characteristics of potential probiotics with significant health benefits [35].

### 2.3. Neurodegenerative Diseases (NDs)

Alzheimer’s disease (AD), Parkinson’s disease (PD), and Huntington’s disease (HD) are the most prevalent in the NDs [7]. Alois Alzheimer first described the ND that would later take his name more than 100 years ago. Even today, the pathological diagnosis of the disease still necessitates the evidence of the amyloid plaques and neurofibrillary tangles he described [36]. The World Alzheimer Report from 2021 estimates about 50 million dementia sufferers worldwide. By 2030 and 2050, this number is expected to increase to around 82 million and 152 million, respectively. There are 200 different types of dementia, and 50–60% of all cases are caused by AD [37,38]. The most prevalent ND in the world is AD [5]. AD refers to a particular onset and progression of cognitive and functional deterioration brought on by aging, ultimately leading to mortality [39]. AD is a chronic and persistent ND that primarily affects the elderly. Clinically, AD is defined by psychological symptoms affecting intellectual and cognitive functions [40]. Even though there have been a lot of studies completed in the last few years about the pathophysiology of AD, its origin remains unknown. Hereditary changes account for less than 5% of cases of dementia, which is primarily an age-related, sporadic disease [38]. The pathogenesis and progression of AD are significantly influenced by neuroinflammation, a process associated with the initiation of several NDs [41]. Neuritic plaques (extracellular deposits of beta-amyloid in brain tissue) and neurofibrillary tangles (intraneuronal deposits of insoluble hyperphosphorylated Tau proteins), which predominate mainly in the hippocampus and temporal cortical regions, are hallmarks of AD [2].

The second most prevalent neurodegenerative condition after AD is PD. PD is a progressive neurological disorder that causes motor impairments due to the degeneration and depletion of dopaminergic neurons in the substantia nigra pars compacta (SNpc) [42]. PD is characterized by chronic tremors, bradykinesia, rigidity, and postural instability. The pathogenic process includes high levels of oxidative stress, mitochondrial malfunction, and apoptosis at the cellular level [43]. Slow movements (bradykinesia) are a primary symptom of PD patients’ hypokinetic movement condition. The most typical symptoms, particularly in the advanced stages of the illness, include body stiffness (hypertonia), tremor, flexed posture, loss of postural reflexes, and freezing. Reduced gait speed and step length, festination, impaired rhythmicity, and increased axial rigidity are the primary gait abnormalities seen in PD patients [44]. PD has generally been studied in the framework of the CNS; however, peripheral effects have also been linked to the development of PD. In fact, new findings indicate that there is bidirectional communication between the brain and the intestines [43].

### 2.4. Recent In Vitro, In Vivo, and Clinical Evidence of Neuroprotective Activities of Probiotics LAB

In vitro studies in this review utilized conditioned medium (CM), LAB-derived exopolysaccharides (EPS) or extracellular vesicles (EVs), as well as direct incubation of LAB in cell culture for the evaluation of LAB neuroprotective activities (Table 1). It was reported that CM of LAB formulations (Lab4 and Lab4b) induced similar levels of proliferation in SH-SY5Y neuroblastoma cells and was able to protect undifferentiated cells from the cytotoxic effect of rotenone and 1-methyl-4-phenylpyridinium (MPP+), as well as differentiated cells from the cytotoxic actions of rotenone. Both formulations attenuate rotenone-induced apoptosis and necrosis and are able to attenuate intracellular ROS accumulation in SH-SY5Y cells. The differential upregulation of genes encoding glutathione reductase and superoxide dismutase by Lab4 CM and Lab4b CM, respectively, indicates different mechanisms of actions exerted by each probiotic consortium [45]. In a study [46], CM was prepared using HT-29 cells cultured with the heat-killed *Lactococcus lactis*, *Lacticaseibacillus rhamnosus* GG, *Lactobacillus delbrueckii* KU200170, and *Lactobacillus plantarum* KU200661. Among the bacterial strains tested, the oxidatively stressed SH-SY5Y cells induced by hydrogen peroxide (H_2_O_2_) were most viable when cultured with *L. lactis* KC24-CM. *L. lactis* KC24-CM enhanced brain-derived neurotropic factor (BDNF) expression in the HT-29 cells. In the oxidatively stressed SH-SY5Y cells, it promotes a significant increase in the expression of BDNF and reduced the apoptosis-related Bax/Bcl-2 ratio.

EPS are long branched-chain polysaccharides, formed by repeating units of carbohydrates or their derivatives [47]. EPS extracted from bacterial cultures (*Lactobacillus delbrueckii* ssp. *Bulgaricus* B3 and *Lactobacillus plantarum* GD2) were incubated with SH-SY5Y and resulted in reduced apoptotic activity exerted by oligomeric amyloid-beta 1-42 (Aβ1-42). The EPS also exhibit a depolarizing effect on mitochondrial membrane potential in a concentration-dependent manner [48]. The protective effect of EPS is assumed to be influenced by its mannose ratio, molecular weight, functional groups, surface morphology, and amorphous character structure [48]. In another study, EPS extracted from similar bacterial cultures protected SH-SY5Y cells from Aβ-mediated neurotoxicity by lowering oxidative stress via maintaining total antioxidant–oxidant status, along with the activities of superoxide dismutase (SOD), catalase (CAT), and glutathione peroxidase (GPx) enzymes. Antioxidative activities of EPS are exerted via upregulation of ERK1, ERK2, JNK, JUN, NF-κB/p65, and p38, as well as by downregulation of AKT/PKB [49].

It was reported that *Lactobacillus plantarum* CRL 1905 (A9 Clone) demonstrated a significant neuroprotective activity resulting in increased neuronal survival of N2a, mouse neuroblastoma cells that were exposed to MPP+ under thiamine-deficient conditions. This result was associated with decreased levels of Interleukin 6 (IL-6) released by the neuronal cells [50]. Heat-killed *Levilactobacillus brevis* KU15152 promoted the expression of brain-derived neurotrophic factor (BDNF) in HT-29 cells. CM of *L. brevis* KU15152 protected neuroblastoma cells (SH-SY5Y) against H_2_O_2_-induced oxidative stress and significantly alleviated morphological changes induced by H_2_O_2_. *L. brevis* KU15152–CM significantly downregulated the Bax/Bcl-2 ratio, and the expression of BDNF and tyrosine hydroxylase (TH) was upregulated in SH-SY5Y cells. Caspase-3 activity was also reduced by *L. brevis* KU15152–CM following the H_2_O_2_ treatment [51]. EVs from *Leuconostoc mesenteroides, Latilactobacillus curvatus*, and *Lactiplantibacillus plantarum* significantly inhibited the production of NO and pro-inflammatory cytokine by LPS-stimulated microglia cells with neuroinflammation. This inhibitory effect is associated with the suppression of the MAPK signaling pathways through phosphorylation of p38 and ERK [52].

**Table 1 pharmaceuticals-16-00712-t001:** In vitro evidence of neuroprotective activities of LAB as probiotics.

Main Author, Year	LAB/Probiotics Formulation Containing LAB	Activities/Results	Ref.
Michael et al., 2019	Formulation Lab4: *Lactobacillus acidophilus* CUL21 (NCIMB 30156) *Lactobacillus acidophilus* CUL60 (NCIMB 30157) *Bifidobacterium bifidum* CUL20 (NCIMB 30153) *Bifidobacterium animalis* subsp. *lactis* CUL34 (NCIMB 30172) Formulation Lab4b: *Lactobacillus salivarius* CUL61 (NCIMB 30211) *Lactobacillus paracasei* CUL08 (NCIMB 30154) *Bifidobacterium bifidum* CUL20 (NCIMB 30153) *Bifidobacterium animalis* subsp. *lactis* CUL34 (NCIMB 30172))	Induced proliferation in SH-SY5Y cells. Protect undifferentiated cells from cytotoxic effect of rotenone and MPP+. Protect differentiated cells from the cytotoxic actions of rotenone. Attenuate intracellular ROS accumulation in SH-SY5Y cells.	[45]
Lim et al., 2020	*Lactococcus lactis* *Lacticaseibacillus rhamnosus* GG *Lactobacillus delbrueckii* KU200170 *Lactobacillus plantarum* KU200661	*L. lactis* KC24-CM protected SH-SY5Y cells against oxidative stress. Enhanced expression of brain-derived neurotropic factor (BDNF). Reduced the apoptosis-related Bax/Bcl-2 ratio.	[46]
Sirin and Aslim, 2020	*Lactobacillus delbrueckii* ssp. *bulgaricus* B3 *Lactobacillus plantarum* GD2	LAB exopolysaccharides reduced the apoptotic activity exerted by Aβ1-42 on SH-SY5Y cells. Exerted depolarizing effect on mitochondrial membrane potential.	[48]
Sirin and Aslim, 2020	*Lactobacillus delbrueckii* ssp. *bulgaricus* B3 *Lactobacillus plantarum* GD2	Exopolysaccharides protected SH-SY5Y cells from Aβ mediated neurotoxicity. Antioxidative activities exerted via upregulation of ERK1, ERK2, JNK, JUN, NF-κB/p65, and p38 and also by downregulation of AKT/PKB.	[49]
Teran et al., 2021	*Lactobacillus plantarum* CRL 1905 *Lactobacillus casei* CRL 238 *Lactobacillus brevis* CRL 2013 *Lactococcus lactis* subsp. *cremoris* CRL 462 and MG1363	Significant neuroprotective activity by *Lactobacillus plantarum* CRL 1905 (A9 clone) associated with decreased levels of IL-6 released by the neuronal cells.	[50]
Bock et al., 2022	*Levilactobacillus brevis* KU15152	*Levilactobacillus brevis* KU15152-CM protected SH-SY5Y cells against H_2_O_2_-induced oxidative stress.	[51]
Kim et al., 2022	*Leuconostoc mesenteroides* *Latilactobacillus curvatus* *Lactiplantibacillus plantarum*	Significantly inhibited production of NO and pro-inflammatory cytokine by neuro-inflamed LPS-stimulated microglia cells.	[52]

In recent years, a number of in vivo studies were carried out to determine the neuroprotective effects of LAB as well as the mechanism involved (Table 2). A study reported that the group of mice that received a *Lactobacillus paracasei*-K71-supplemented diet had better cognitive performance in the Barnes maze and passive avoidance tests compared to the control group. Serotonin levels in the serum and brain tissue of *L. paracasei*-K71-fed mice were elevated. Long-term supplementation of *L. paracasei* K71 resulted in upregulation in protein expression of BDNF and CREB phosphorylation in the hippocampus [53]. LAB probiotic (VSL#3^®^) supplementation in aflatoxin-B1-induced neurotoxicity in adult male rats resulted in decreased oxidative stress and inflammatory markers. The probiotic supplementation is effective at reducing the neurotoxic effects of aflatoxin B1 in rats [54]. Another LAB probiotic formulation known as ProBiotic-4 significantly improved the memory deficits, cerebral neuronal and synaptic injuries, glial activation, and microbiota composition in the feces and brains of aged SAMP8 mice. Its mechanism of action is associated with the inhibition of both TLR4- and RIG-I-mediated NFkB signaling pathways as well as inflammatory responses [55]. Treatment with *Lacticaseibacillus rhamnosus*, *Lactobacillus reuteri*, or *Lactobacillus plantarum* significantly prevented LPS-induced elevation of TNF-α mRNA and memory deterioration in rats. EV from *Lactobacillus plantarum* has demonstrated significant antidepressant-like activity in mice by elevating BDNF expression [56]. *Lacticaseibacillus rhamnosus* prevented changes induced by LPS on CaMKII-α mRNA levels. These bacteria strains also exert neuroprotective activity via their immune-modulatory properties [57].

There are seven in vivo studies included in this review which utilized AD animal models to investigate LAB neuroprotective activities. *Lactobacillus acidophilus* supplementation resulted in significant changes in neurotransmitters and antioxidant levels in AD-induced rats. Reduced ATP1A1 activity and MMP gene expression in the hippocampus were detected when compared to the control. *Lactobacillus acidophilus* was able to alleviate learning- and memory-associated injuries in Alzheimer’s rats by reducing mitochondrial dysfunction induced by D-gal and aluminum chloride (AlCl3). This could be associated with its antioxidant properties [58]. Probiotics formulation containing LAB (*Lactobacillus acidophilus* (1688FL431-16LA02), *Lactobacillus fermentum* (ME3), *Bifidobacterium lactis* (1195SL609-16BS01), and *Bifidobacterium longum* (1152SL593-16BL03)) was administered to AD-induced rats and resulted in significant improvement in spatial memory, including scape latency, travelled distance, and time spent in the target quadrant. Oxidative stress biomarkers were also improved such as elevated malondialdehyde levels and superoxide dismutase activity following the β-amyloid (Aβ) injection [59]. Supplementation of *Lactobacillus acidophilus*, *Bifidobacterium bifidum*, and *Bifidobacterium longum* helped to improve the maze navigation and significantly restored long-term potentiation (LTP) in rats [60]. The same probiotic strains also helped to significantly improve learning, but not memory impairment, and also increased paired-pulse facilitation (PPF) ratios in AD rats [61].

In a study, mice were treated with SLAB51 (LAB probiotic formulation) for 16 weeks, and the effects on protein oxidation, neuronal antioxidant defense, and repair systems were evaluated, focusing on the role of SIRT1-related pathways. It was found that SLAB51 significantly reduced oxidative stress in AD mice brains by activating SIRT1-dependent mechanisms [62]. Another study used the same probiotic formulation as supplementation in AD mice which resulted in an improvement in glucose uptake by restoring the levels of key glucose transporters (GLUT3 and GLUT1) and insulin-like growth factor receptor b in the brain, in conformity with the diminished phosphorylation of adenosine-monophosphate-activated protein kinase and protein kinase B (Akt). SLAB51 hinders the time-dependent increase in glycated hemoglobin and the accumulation of advanced glycation end products in AD mice, which is consistent with memory improvement [63]. Probiotic VSL#3 supplementation altered levels of lactate and acetate in both serum and brain of AppNL-G-F mice correlating with upregulated expression of the neuronal activity marker, c-Fos [64]. Probiotics (*Lactobacillus reuteri*, *Lacticaseibacillus rhamnosus*, and *Bifidobacterium infantis*) supplemented AD rats had significant improvement in spatial memory and reduced Aβ plaques. These strains also helped to reduce IL-1β and TNF-α inflammation markers in the AD model of rats [65].

Four studies investigated the effects of LAB on PD animal models. In a study, PD mice were treated with bacteria strains (*Lactobacillus plantarum* CRL 2130, *Streptococcus thermophilus* CRL 807, and *Streptococcus thermophilus* CRL 808) individually and as a mixture. LAB supplementation resulted in significantly enhanced motor skills. The mice that received a mixture of all three LAB strains had higher brain TH-positive cell counts, reduced inflammatory cytokines IL-6 and TNF-α in serum, as well as elevated levels of anti-inflammatory cytokine IL-10 in serum and brain tissues, which resulted in improvement in neuroinflammation [43]. Moreover, it was suggested that the particular LAB mixture may reduce PD-related neuroinflammation and motor behavior problems. Therefore, this probiotic combination may be potentially used as an additional form of treatment for PD [43]. Another research has shown that LAB decreased pro-inflammatory cytokines in serum [42]. In addition, the LAB strain that has higher thiamine production demonstrated the most potent anti-inflammatory effect locally in the brain. It significantly decreased the levels of IL-6, TNF-α, IFN-, and MCP-1. These results demonstrated the efficacy of *Lactiplantibacillus plantarum* CRL 1905, a thiamine-producing strain, as a strategy to prevent or augment the treatment of neurological disorders. Furthermore, recent research has shown that the treatment by a combination of LAB chosen for their anti-inflammatory and antioxidant properties (including a strain that produces riboflavin) enhanced motor skills in a Parkinsonian mouse model [42].

**Table 2 pharmaceuticals-16-00712-t002:** In vivo evidence of neuroprotective activities of LAB as probiotics.

Main Author, Year	LAB/Probiotic Formulation Containing LAB	Animal	Activities/Results	Ref.
Azm et al., 2018	*Lactobacillus acidophilus* (1688FL431-16LA02) *Lactobacillus fermentum* (ME3) *Bifidobacterium lactis* (1195SL609-16BS01) *Bifidobacterium longum* (1152SL593-16BL03)	Eight-week-old male Wistar rats Aβ-induced Alzheimer’s disease (AD) model	Significant improvement in spatial memory in rats. Improved oxidative stress biomarkers.	[59]
Bonfili et al., 2018	Formulation SLAB51: *Lactobacillus acidophilus* DSM 32241 *Lactobacillus helveticus* DSM 32242 *Lactobacillus paracasei* DSM 32243 *Lactobacillus plantarum* DSM 32244 *Lactobacillus brevis* DSM 27961 *Bifidobacterium lactis* DSM 32246 *Bifidobacterium lactis* DSM 32247 *Streptococcus thermophilus* DSM 32245	Eight-week-old male transgenic Alzheimer’s disease (AD) mice (3xTg-AD)	Significantly reduced oxidative stress in AD mice brain by activating SIRT1-dependent mechanisms.	[62]
Corpuz et al., 2018	*Lactobacillus paracasei* K71 *Lactobacillus casei* subsp. *Casei zz*	Fourteen-week-old female SAMP8 mice	Better cognitive performance in the Barnes maze and passive avoidance tests after K71 supplementation. Upregulation in BDNF expression and CREB phosphorylation in the hippocampus.	[53]
Choi et al., 2019	*Lactobacillus plantarum*	Seven-week-old male C57BL/6J mice	Reduced depression-like behavior by elevating BDNF expression.	[56]
Rezaei Asl et al., 2019	*Lactobacillus acidophilus* *Bifidobacterium bifidum* *Bifidobacterium longum*	Adult male Wistar rats Aβ-induced Alzheimer’s disease (AD) model	Improved maze navigation. Restored long-term potentiation (LTP) in rats.	[60]
Rezaeiasl et al., 2019	*Lactobacillus acidophilus* *Bifidobacterium bifidum* *Bifidobacterium longum*	Adult male Wistar rats Aβ-induced Alzheimer’s disease (AD) model	Significant improvement in learning. Increased paired-pulse facilitation (PPF) ratios.	[61]
Bonfili et al., 2020	Formulation SLAB51: *Lactobacillus acidophilus* DSM 32241 *Lactobacillus helveticus* DSM 32242 *Lactobacillus paracasei* DSM 32243 *Lactobacillus plantarum* DSM 32244 *Lactobacillus brevis* DSM 27961 *Bifidobacterium lactis* DSM 32246 *Bifidobacterium lactis* DSM 32247 *Streptococcus thermophilus* DSM 32245	Eight-week-old male transgenic Alzheimer’s disease (AD) mice (3xTg-AD)	Improvement in glucose uptake in the brain. Memory improvement.	[63]
Mehrabadi and Sadr, 2020	*Lactobacillus reuteri* *Lacticaseibacillus rhamnosus* *Bifidobacterium infantis*	Male Wistar rats, Aβ1-40-induced Alzheimer’s disease (AD) model	Significant improvement in spatial memory and reduced Aβ plaques. Reduced IL-1β and TNF-α inflammation markers.	[65]
Kaur et al., 2020	VSL#3^®^ Formulation: *Lactobacillus acidophilus* *Lactobacillus plantarum* *Lactobacillus paracasei* *Lactobacillus delbrueckii* subsp. *bulgaricus* *Streptococcus thermophilus* *Bifidobacterium longum* *Bifidobacterium breve* *Bifidobacterium infantis*	C57BL/6 wild-type (WT) App^NL-G-F^ mice, Alzheimer’s disease (AD) model	Altered levels of lactate and acetate in both serum and brain. Upregulated expression of the neuronal activity marker, c-Fos.	[64]
Visnuk et al., 2020	*Lactobacillus plantarum* CRL 2130 (a riboflavin producer) *Streptococcus thermophilus* CRL 807 (an immunomodulatory strain) *Streptococcus thermophilus* CRL 808 (a folate producer)	Eight-week-old male C57BL/6 mice (MPTP/probenecid parkinsonism model)	Enhanced motor skills. Higher brain tyrosine-hydrolase-positive cell counts. Reduced IL-6 and TNF-α in serum. Elevated levels of IL-10 in serum and brain tissues. Improvement in neuroinflammation.	[43]
Yang et al., 2020	ProBiotic-4 formulation: *Lactobacillus casei* *Lactobacillus acidophilus* *Bifidobacterium lactis* *Bifidobacterium bifidum*	Nine-month-old senescence-accelerated mouse prone 8 (SAMP8)	Improved the memory deficits, cerebral neuronal and synaptic injuries, glial activation, and microbiota composition in the feces and brains.	[55]
Beltagy et al., 2021	*Lactobacillus acidophilus*	Adult male albino rats (6–7 weeks age)	Significant changes in neurotransmitters and antioxidants levels. Alleviate learning- and memory-associated injuries in Alzheimer’s rats.	[58]
Wang et al., 2021	*Lactobacillus plantarum* DP189	MPTP-induced Parkinson’s disease (PD) model mice	Improved behavioral ability. Significant neuroprotective effect in PD mice.	[66]
Zolfaghari et al., 2021	*Lacticaseibacillus rhamnosus* *Lactobacillus reuteri* *Lactobacillus plantarum*	Adult male Wistar rats	Prevented LPS-induced elevation of TNF-α mRNA and memory deterioration. Probiotics exert neuroprotective activity via immune-modulatory properties.	[57]
Liu et al., 2022	*Lacticaseibacillus rhamnosus* GG	Male C57BL/6J mice (8-week-old)	Prevented dopaminergic neuronal loss and weakened muscle strength in behavior tests. Neuroprotective activities possibly due to enhancement of striatal glial-cell-derived neurotrophic factor (GDNF) expression.	[67]
Sahin et al., 2022	VSL#3^®^ Formulation: *Lactobacillus acidophilus* *Lactobacillus plantarum* *Lactobacillus paracasei* *Lactobacillus delbrueckii* subsp. *bulgaricus* *Streptococcus thermophilus* *Bifidobacterium longum* *Bifidobacterium breve* *Bifidobacterium infantis*	Eight-week-old Wistar Albino rats	Decreased oxidative stress and inflammatory markers. Reduced neurotoxic effects of aflatoxin B1 in rats.	[54]
Visnuk et al., 2022	*Lactiplantibacillus plantarum* CRL2130 *Lactiplantibacillus plantarum* CRL725	Eight-week-old male C57BL/6 mice (MPTP/probenecid parkinsonism model)	Attenuated motor deficits Prevented dopaminergic neuronal death. Decreased pro-inflammatory cytokines and increase in IL-10 levels in serum and brain	[42]

One study reported that mice treated with *Lactobacillus plantarum* DP189 demonstrated improved behavioral ability as well as increased levels of 5-hydroxytryptamine and dopamine. The positive rate of TH cells was significantly increased after the treatment. *L. plantarum* DP189 activated the ERK2 and AKT/mTOR pathways, enhanced the expression of Bcl-2, and inhibited Bax and Caspase 3 activities. The treatment resulted in significant neuroprotective effects in PD mice [66]. Chronic PD mice that received *Lacticaseibacillus rhamnosus* GG or prebiotic polymannuronic acid, separately or in combination for 5 weeks, were protected from dopaminergic neuronal loss and weakened muscle strength in behavior tests by enhancement of the TH gene in the midbrain and striatum. The neuroprotective activities of *L. rhamnosus* GG are possibly due to its ability to enhance the expression of the striatal glial-cell-derived neurotrophic factor (GDNF) [67].

There are two clinical studies included in this review that focused on the effects of LAB probiotics alone or in combination towards the elderly and patients with cognitive impairments (Table 3). A study reported that the group of patients diagnosed with mild cognitive impairment (MCI) who received *Lactobacillus plantarum* combined with fermented soybean powder supplementation had shown greater improvements in the combined cognitive functions, especially in the attention domain. These improvements are associated with elevated serum BDNF levels [68]. In addition, co-supplementation of probiotic *Lactobacillus acidophilus*, *Bifidobacterium bifidum*, and *Bifidobacterium longum* with selenium to AD patients resulted in favorable effects on Mini-Mental State Examination (MMSE) score as well as improved levels of hs-CRP, TAC, GSH, insulin metabolism markers, triglycerides, VLDL-, LDL-, and total-/HDL cholesterol. No effect was demonstrated on other inflammation biomarkers and oxidative stress, FPG and other lipid profiles. The co-supplementation enhanced the gene expression of TNF-a, PPAR-g and LDLR, but had no effect on the gene expression of IL-8 and TGF-b. Cognitive function and some metabolic profiles were also improved [69].

**Table 3 pharmaceuticals-16-00712-t003:** Clinical evidence of neuroprotective activities of LAB as probiotics.

Main Author, Year, Country, Study Design	LAB/Probiotic Formulation Containing LAB	Participants	Results	Ref.
Hwang et al., 2019, South Korea, Multi-center, randomized, double-blind, controlled clinical trial	*Lactobacillus plantarum* combined with fermented soybean powder	100 men and women diagnosed with mild cognitive impairment (MCI) according to the Diagnostic and Statistical Manual of Mental Disorders, 5th edition (DSM-5), were recruited at Chonbuk National University Hospital and Kyung Hee University Hospital	Greater improvements in the combined cognitive functions. Elevated serum BDNF levels.	[68]
Tamtaji et al., 2019, Iran, Randomized, double-blind, controlled trial	*Lactobacillus acidophilus* *Bifidobacterium bifidum Bifidobacterium longum* co-supplemented with selenium	AD patients (55–100 years of age) at the Golabchi Welfare Organization (Kashan, Iran) and Madar, Shayestegan, and Amin Welfare Organizations (Shahrekord, Iran)	Improved MMSE score, hs-CRP, TAC, GSH, insulin metabolism markers, triglycerides, VLDL-, LDL-, total-/HDL cholesterol. Enhanced gene expression of TNF-a, PPAR-g, and LDLR. Improved cognitive function and some metabolic profiles.	[69]

### 2.5. Mechanisms of Action of LAB on Neuroprotective Activity

Oxidative stress is a major contributor to the progression of several diseases, including aging, inflammatory, and neurodegenerative conditions [70]. It has been proposed that LAB may have antioxidant activities through scavenging ROS, chelating metals, boosting antioxidant enzyme levels, and regulating the microbiota [70]. In addition, oxidative stress causes neural cell damage. In this context, antioxidants have the ability to protect cells from oxidative-stress-induced damage [71]. 

Superoxide dismutase (SOD), catalase (CAT), and glutathione peroxydase (GSH-Px) are the most important endogenous antioxidant enzymes. SOD converts superoxide anion to H_2_O_2_, which is a substrate for CAT and GSH-Px. CAT metabolizes H_2_O_2_ in water and oxygen, and GSH-Px reduces both H_2_O_2_ and organic hydroperoxydes when reacting with glutathione (GSH) [45,72]. Furthermore, GSH is also a well-known indicator of oxidative stress and cellular detoxification [54]. In vivo experiments found that LAB has been shown to play a role in the antioxidant effect demonstrated in the AD model, which markedly attenuates ROS levels by increased levels of GSH and SOD activity [62]. In another animal model, there was an increase in GSH level and GSH-Px activity, leading to antioxidant activities in the probiotic-treatment group [54]. In addition, LAB increased the SOD and decreased the MDA enzyme level, indicating that LAB has an antioxidative effect [65]. Moreover, the efficacy of *Lactiplantibacillus plantarum* CRL 1905, a thiamine-producing LAB, can protect against oxidative stress via an indirect effect, possibly mediated by an antioxidant signaling pathway, rather than a direct ROS scavenging effect [43,50]. Similar results were reported that combined probiotic and selenium supplementation may improve AD patients by correcting metabolic abnormalities and attenuating inflammation and oxidative stress by increasing the antioxidant marker of GSH [69].

LAB also exerts immunostimulatory effects as demonstrated in macrophage cells and T-lymphocytes isolated from the spleen of mice [52]. Tumor necrosis factor-alpha (TNF-α), interleukin-1 beta (IL-1β), and nuclear factor-kappa B (NF-KB) are proteins that are directly involved in neuroinflammatory processes and the activation of microglia [73]. In addition, TNF-α is a proinflammatory cytokine important in initiating and stimulating inflammatory responses [57]. As a result of the foregoing, microglial cells play a critical role in innate immunity and are the primary source of proinflammatory factors in the human brain. The immune-modulatory properties of the LAB have been demonstrated to influence brain function and memory performance [57]. Previous studies have revealed that IL-1β and TNF-α as inflammation markers in the AD model of rats were reduced through supplemented with LAB [65]. In addition, it has been reported that consuming LAB significantly reduced the effects of LPS on TNF-mRNA expression and CaMKII-mRNA levels that contribute to memory deterioration. Moreover, it was suggested that LAB treatment might reduce motor deficits and prevent dopaminergic neuronal death, with a decrease in proinflammatory cytokines and an increase in IL-10 in both serum and brain compared to the control group (without treatment) [42].

Acetylcholinesterase (AChE), which is found in synaptic gaps, is responsible for acetylcholine breakdown and is thus thought to play a role in the development of AD [74]. The AChE-inhibitory activity of LAB has been shown to have neuroprotective properties by reducing the amount and formation of Aβ plaques as a therapeutic approach to AD. In addition, acetylcholine is essential for cognitive function, and AChE inhibitors may increase acetylcholine levels. As a result, AChE inhibitors are another effective anti-AD therapeutic strategy [75]. A study reported that LAB administration might improve AChE and acetylcholine concentrations in Alzheimer’s disease patients [58].

LAB has also demonstrated neuroprotective activities by antiapoptotic strategy. When apoptosis is induced, the pro-apoptotic BCL-2 family protein known as BAX is translocated from the cytosol to the mitochondria. The BAX/BCL-2 ratio has a greater impact on a cell’s resistance to apoptosis compared to the expression of each gene [46]. A decrease in apoptosis can be explained by a reduced BAX/BCL-2 ratio. In order to maintain the integrity of the mitochondrial outer membrane, antiapoptotic BCL-2 proteins primarily act as inhibitors of pro-apoptotic BAX [76]. A few studies reported that treatment by LAB *Lactococcus lactis* [46], *Lacticaseibacillus rhamnosus GG* [67], and *Levilactobacillus brevis* [51] has downregulated BAX/BCL-2 ratio, thus inhibiting apoptosis and exerting neuroprotective effects.

A neurotrophic factor that supports the differentiation, maturation, and survival of neurons in the nervous system is called brain-derived neurotrophic factor (BDNF), which also has neuroprotective effects in the presence of adverse conditions such as neurotoxicity [77]. BDNF has a main role in the enhancement of cognitive function using molecules involved in synaptic plasticity and cognitive processes [78]. The transcription factor CREB and other intracellular signaling pathways closely control the transcription of BDNF and other neurotrophins [53]. Studies reported that LAB treatments helped in upregulation of BDNF expression in neuronal cells [51] and increased BDNF serum level [68]. It was suggested that the increase in BDNF expression brought on by long-term supplementation with LAB (L. K71) may be mediated through CREB which helped to maintain neural plasticity and brain function [53]. The function of BDNF is negatively affected by Aβ. Aβ peptide is a critical initiator that triggers the progression of AD via accumulation and aggregation [79]. Probiotic formulation containing LAB may enhance synaptic transmission in the hippocampal formation by removing Aβ plaques or preventing their development [60]. EPS extracted from *Lactobacillus delbrueckii* ssp. *Bulgaricus* B3 and *Lactobacillus plantarum* GD2 directly inhibit fibril formation by Aβ, showing antiaggregative activity of the LAB-derived EPS [48].

Extracellular vesicles (EVs), also known as membrane vesicles, are lipid-bilayer-enclosed particles with potential therapeutic use [80]. EVs produced by bacteria, notably LAB, are used in intercellular communication between the microbiota and the host. EVs transport lipids, proteins, bacterial metabolites, and nucleic acids that can affect the host’s cellular pathways [56]. EVs derived from *Lactobacillus plantarum* change the expression of neurotropic factors in the hippocampus, and SIRT1 was revealed to play a role in EV-induced upregulation of BDNF and CREB expression [56]. SIRT1 is an NAD+-dependent deacetylase that has shown to protect neurons in various in vitro and in vivo models of neurodegenerative diseases [81]. It is widely recognized that reduced SIRT1 expression has negative effects, including the buildup of Aβ and tau in AD patients’ cerebral cortex [82]. The probiotic formulation containing LAB (SLAB51) significantly elevated SIRT1 expression and activity in the brains of AD mice [62]. Untreated AD mice showed an age-dependent rise in the degree of acetylation, which was consistent with the decreased expression of SIRT1. SLAB51 significantly decreased the level of RARβ-acetylated lysines by increasing SIRT1 levels [62]. It is interesting to note that activating RARβ stimulates transcription of the ADAM10 gene, promoting the nonamyloidogenic pathway of APP processing and inhibiting the production and deposition of Aβ peptides [83].

p38 MAPK signaling is crucial for controlling cellular functions, especially inflammation [84]. A crucial component of the neuroinflammatory system set off by glial cells during the onset of neurodegenerative disorders is the MAP-ERK pathway [85]. EVs derived from LAB (*Leuconostoc mesenteroides*, *Latilactobacillus curvatus*, and *Lactiplantibacillus plantarum*) demonstrated anti-inflammatory activity on microglial cells, mediated by effects on p38 and ERK signaling pathways [52]. Pretreatment of LAB-derived EVs prior to LPS induction has significantly decreased the activation of phosphorylated ERK and p38 protein in comparison to control LPS-treated microglial cells [52].

Although MAPK signaling controls cellular survival, differentiation, and proliferation, AMPK signaling governs cellular metabolism [86]. Due to its capacity to phosphorylate Tau, AMPK plays a significant role in the pathogenesis of AD. Together with increased Tau phosphorylation, p-AMPK also contributes to the production and buildup of Aβ [87]. An early AD hallmark that results in cognitive dysfunction is low glucose absorption or a decrease in brain glucose levels [88]. In the aged AD animals’ brains, GLUT levels declined while phosphorylated AMPK and Akt levels increased. These kinases control the expression of GLUTs and are essential for glucose uptake. By restoring the level of GLUT expression in the brain and reducing the phosphorylation of important metabolic regulators such as AMPK and Akt, LAB probiotics formulation can tackle insulin resistance and, as a result, lower Tau phosphorylation [62]. It was suggested that *Lactobacillus plantarum* DP189 may induce the phosphorylation of Akt protein and upregulate the mTOR level by increasing the production of ERK2, which further facilitates the phosphorylation of mTOR and subsequently exerts a neuroprotective effect in dopaminergic neurons [66].

A four-carbon nonprotein amino acid known as GABA (γ-aminobutyric acid) is widely distributed in animals, plants, and microorganisms including LAB [89]. The physiological roles of GABA are related to the modulation of synaptic transmission as well as the enhancement of neuronal development [90]. GABA synthesized by an LAB strain, *Lactobacillus buchneri* was reported to demonstrate neuroprotective activity towards neuronal cells (PC12) against neurotoxicant-induced cell death [91]. GABA levels in the intestines and the central nervous system have been demonstrated to reduce when *Lactobacillus* levels decline, and this correlates with the inhibition of an inflammatory immune response. The increased neuronal excitability observed in LAB probiotic formulation (VSL#3) treated App NL-G-F mice may also be linked to altered levels of neurotransmitters in the brain [64].

As a summary, the proposed modes of action of LAB on neuroprotective activity are illustrated in Figure 2. From this review, there is a number of evidence showing LAB exerts its neuroprotective activities mainly via antioxidant and anti-inflammatory mechanisms. By elevating the expression of GSH, GSH-Px, and SOD, LAB protects neuronal cells via antioxidant pathways. LAB also secretes EVs that tackle neuroinflammation by altering pro-inflammatory and anti-inflammatory cytokines expression in the brain. LAB-derived EVs and EPSs play important roles in the prevention of Aβ plaque formation as well as accumulation via antiaggregative activities. AChE inhibition is also one of LAB mechanisms of action that help to improve acetylcholine level in the brain. LAB also targets AMPK and Akt signaling pathways which eventually reduced Aβ formation. EVs and GABA produced by LAB have a significant impact on synaptic plasticity and neuronal excitability, respectively. Finally, LAB exerts neuroprotection by antiapoptotic strategy via regulation of apoptosis genes, BAX/BCL-2 ratio.

## 3. Methods

Three electronic databases, namely, Google Scholar, PubMed and Science Direct, were searched for articles that evaluated and reported the neuroprotective activity of LAB or the effect of LAB on neurodegenerative diseases between 2018 and 2023. For PubMed and Science Direct, the search strategy was using keywords “lactic acid bacteria” AND “probiotic” AND “neuroprotective” OR “neuroprotection” OR “neurodegenerative” OR “Alzheimer’s” OR “Parkinson’s”. For Google Scholar, a search feature “Advance Search” was applied to find articles with all the words “lactobacillus neuroprotective neuroprotection”, with the exact phrase “lactic acid bacteria”, with at least one of the words “neurodegenerative Alzheimer’s Parkinson’s neuroprotective neuroprotection”, and without the word “systematic review meta-analysis”. Reference lists were also screened for relevant topics as an additional source of literature search.

This review includes in vitro, in vivo, and clinical studies that focused on the effects of lactic acid bacteria as probiotics on neuroprotection. Only research with full text and English language from years 2018 to 2023 were selected. Review articles and meta-analysis studies were excluded from the selection. Studies on lactic acid bacteria probiotics activities that are unrelated to neuroprotection or neurodegenerative diseases were excluded from this review. Extraction of data encompassing main author, year, country, study design, probiotics, animals, participants, as well as results or outcomes of study was performed by two authors, and the findings were discussed in this review.

## 4. Conclusions

In summary, this review provided insights into the neuroprotective potential of LAB as probiotics based on recent in vitro, in vivo, and clinical studies. Antioxidant and anti-inflammatory activities of LAB appeared to be the main mechanisms of protection against neurotoxicity or in alleviating symptoms of neurodegenerative diseases. In future, it is recommended that a more comprehensive analysis of clinical studies, for example, meta-analysis studies be carried out for stronger evidence of LAB neuroprotective activities in humans. Such analysis cannot be performed in this review due to the limited number of studies available.

## Figures and Tables

**Figure 1 pharmaceuticals-16-00712-f001:**
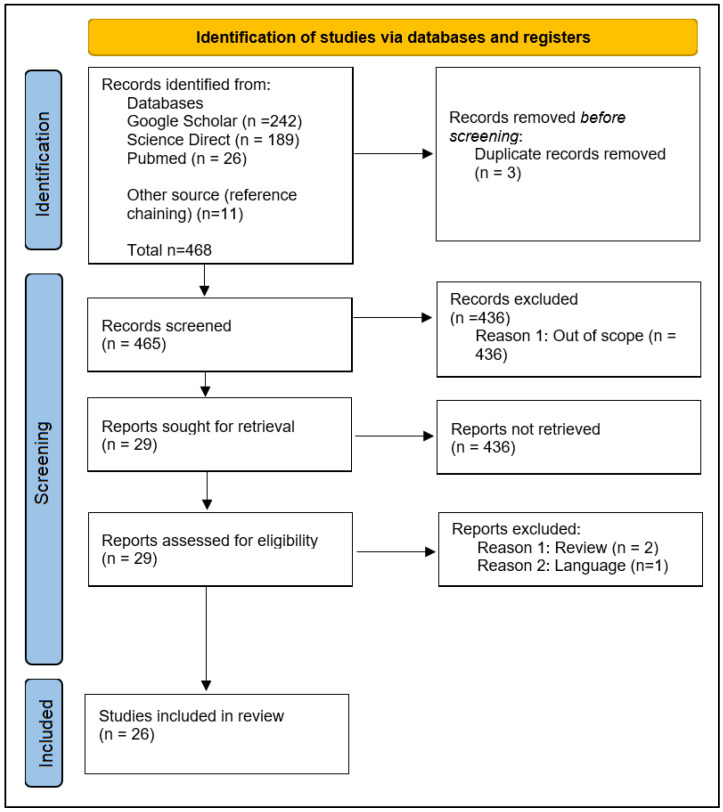
The flowchart of identification and selection of study.

**Figure 2 pharmaceuticals-16-00712-f002:**
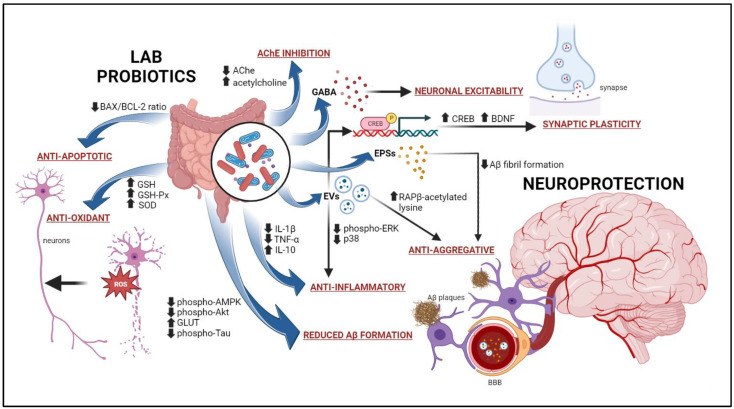
The proposed mechanisms of action of LAB as a neuroprotective agent.

## Data Availability

Data sharing not applicable.

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
