# Peer review of "Lactic Acid Bacteria (LAB) and Neuroprotection, What Is New? An Up-To-Date Systematic Review"

_pharmaceuticals, 2023, doi:10.3390/ph16050712_

Round 1
Reviewer 1 Report
The manuscript is about the neuroprotective properties of Lactic Acid Bacteria. The effect shown by the authors is further evidence of the important role played by LAB including probiotics in normal human functioning. Nevertheless, it would be a valuable addition to show that the relationship between neurodegenerative diseases, general health, the popularity of consuming fermented foods in diverse cultures. Are there fewer neurodegenerative diseases reported in countries/regions where fermented foods containing LAB are a component of the daily diet? I think the authors should consider supplementing the manuscript with these issues.
Some specific comments:
1. please use et al. (with dot)
2. Line 130 - double space
3. Some bacterial names are not actual in the currently accepted taxonomy, for example Lactobacillus rhamnosus is now Lacticaseibacillus rhamnosus, please control all used names.
Author Response
1. We have added citations to show the relationship between memory enhancement and consumption of fermented foods in two countries, Indonesia and Japan. (line 75-80)
2. The use of 'et al' was corrected to 'et al.' throughout the manuscript.
3. Lactobacillus rhamnosus has been changed to Lacticaseibacillus rhamnosus and standardized throughout the manuscript.
Reviewer 2 Report
The manuscript explores the mechanisms underlying the neuroprotective effects of LAB, such as modulating the gut-brain axis, reducing inflammation and oxidative stress, and improving neurotransmitter function. Additionally, the authors discuss the potential therapeutic applications of LAB for neurodegenerative diseases, specifically Alzheimer's and Parkinson's. However, the manuscript has significant shortcomings as the authors list the studies conducted and the obtained results but do not analyze the molecular mechanisms underlying the neuroprotective effect of LAB. The main purpose of a review article is to summarize and describe the major findings published previously, which is not achieved in this manuscript, as it is presented in a brief paragraph form.
To rectify these shortcomings, the authors must significantly revise the manuscript by outlining strategies for using LAB for neuroprotection based on the molecular mechanisms of action on host cells and tissues. The authors also need to cover studies on the neuroprotective properties of LAB-derived extracellular vesicles. The authors write that "It has been proposed that LAB may have antioxidant activities through scavenging ROS, chelating metals boosting antioxidant enzyme levels, and regulating the microbiota", but they should specify the specific molecular mechanisms involved. Furthermore, the authors should conduct analytical work and highlight the primary mechanisms and molecular intermediates of the neuroprotective action of LAB, encompassing the effects of the conditioned medium and EPS.
Author Response
We have further elaborate in section 2.4 on the strategies and pathways involved in using LAB for neuroprotection based on the molecular mechanisms of action on host's cells and tissues targets.
We have included studies on the neuroprotective properties of LAB-derived extracellular vesicles. The main mechanisms and molecular intermediates of the neuroprotective action of LAB have been highlighted in section 2.4 and summarized in Figure 2.
Reviewer 3 Report
Dear authors,
The review article on "Lactic Acid Bacteria (LAB) and Neuroprotection, What’s New?: An Up-to-Date Systematic Review", is a good interesting topic, no one has discussed so for now.
1. Introduction part is written well with proper references.
2. Authors can cite
a. Gallic acid protects 6-OHDA induced neurotoxicity by attenuating oxidative stress in human dopaminergic cell line
b. Gallic Acid from Terminalia Bellirica Fruit Exerts Antidepressant-like Activity
I suggest to the editor, the review paper can be publishable
Author Response
We have added two citations as recommended by the reviewer. (line 62)
a. Gallic acid protects 6-OHDA induced neurotoxicity by attenuating oxidative stress in human dopaminergic cell line
b. Gallic Acid from Terminalia Bellirica Fruit Exerts Antidepressant-like Activity
Round 2
Reviewer 2 Report
Dear Authors,
I am writing to inform you that I have received your revised manuscript for ‘Lactic Acid Bacteria (LAB) and Neuroprotection, What’s New?”: An Up-to-Date Systematic Review, and I am pleased to say that you have done an excellent job of addressing all of my comments and recommendations.
Thank you again for taking the time to address my comments and recommendations.